# Investigation of the Membrane Fluidity Regulation of Fatty Acid Intracellular Distribution by Fluorescence Lifetime Imaging of Novel Polarity Sensitive Fluorescent Derivatives

**DOI:** 10.3390/ijms22063106

**Published:** 2021-03-18

**Authors:** Giada Bianchetti, Salome Azoulay-Ginsburg, Nimrod Yosef Keshet-Levy, Aviv Malka, Sofia Zilber, Edward E. Korshin, Shlomo Sasson, Marco De Spirito, Arie Gruzman, Giuseppe Maulucci

**Affiliations:** 1Neuroscience Department, Biophysics Section, Università Cattolica del Sacro Cuore, 00168 Rome, Italy; giada.bianchetti@unicatt.it (G.B.); marco.despirito@unicatt.it (M.D.S.); 2Fondazione Policlinico Universitario “A. Gemelli” IRCSS, 00168 Rome, Italy; 3Department of Chemistry, Bar-Ilan University, Ramat Gan 59290002, Israel; salomeazoulay16@gmail.com (S.A.-G.); labpath@szmc.org.il (N.Y.K.-L.); aviv1m@walla.co.il (A.M.); edward.korshin@gmail.com (E.E.K.); 4Department of Pathology, Shaare Zedek Medical Center, Jerusalem 9103102, Israel; sofiazilber@szmc.org.il; 5Institute for Drug Research, Faculty of Medicine, The Hebrew University, Jerusalem 911210, Israel; shlomo.sasson@mail.huji.ac.il

**Keywords:** fatty acids analogs, laurdan derivatives, FLIM, membrane fluidity, two-photons microscopy, phasor analysis

## Abstract

Free fatty acids are essential structural components of the cell, and their intracellular distribution and effects on membrane organelles have crucial roles in regulating the metabolism, development, and cell cycle of most cell types. Here we engineered novel fluorescent, polarity-sensitive fatty acid derivatives, with the fatty acid aliphatic chain of increasing length (from 12 to 18 carbons). As in the laurdan probe, the lipophilic acyl tail is connected to the environmentally sensitive dimethylaminonaphthalene moiety. The fluorescence lifetime imaging analysis allowed us to monitor the intracellular distribution of the free fatty acids within the cell, and to simultaneously examine how the fluidity and the microviscosity of the membrane environment influence their localization. Each of these probes can thus be used to investigate the membrane fluidity regulation of the correspondent fatty acid intracellular distribution. We observed that, in PC-12 cells, fluorescent sensitive fatty acid derivatives with increased chain length compartmentalize more preferentially in the fluid regions, characterized by a low microviscosity. Moreover, fatty acid derivatives with the longest chain compartmentalize in lipid droplets and lysosomes with characteristic lifetimes, thus making these probes a promising tool for monitoring lipophagy and related events.

## 1. Introduction

Free fatty acids (FFAs) are essential structural components of the cell and play important roles in intracellular energy metabolism and signaling cascades. Biologically, FFA are esterified with glycerol, phosphoglycerol, and cholesterol and are referred to as triacylglycerol, phospholipids, and cholesteryl esters, respectively. Esterified fatty acids can constitute the structural components or dietary fuels for cells and organisms; they can also form complex liposomal structures (including lipoproteins) for transporting lipid components from the hepatic tissues to extrahepatic tissues and vice versa. FA whose aliphatic carbon chains are fully saturated with hydrogen atoms or contain only C-C single bond are referred to as saturated fatty acids (SFAs). Fatty acids containing single or multiple C=C double bonds are termed unsaturated fatty acids (UFAs). The aliphatic chains and their length confer hydrophobicity to FA, thus rendering them insoluble in aqueous environments. The melting point of fatty acid is inversely related to chain length and it is further decreased by the double bonds in the unsaturated fatty acids [1]. Current views on structural and dynamical aspects of biological membranes have been strongly influenced by the homogenous fluid mosaic model proposed by Singer and Nicolson in 1972 [2]. In 1997, it was hypothesized the existence of lipid rafts [3], defined as small (20–100 nm), heterogeneous, highly dynamic, sterol- and sphingolipid (SL)-enriched domains that compartmentalize cellular processes and are governed by the liquid-ordered (Lo)/liquid-disordered (Ld) phase partitioning described in purified lipid systems. However, the notion of these transient nanometric domains is still insufficient to explain the cellular control of surface lipid diversity or membrane deformability [4]. During the past decade, the hypothesis that some lipids form large (submicrometric/mesoscale vs. nanometric rafts) and stable (>min vs. sec) membrane domains has emerged, based mainly on indirect methods. Morphological evidence for stable submicrometric lipid domains, well-accepted for artificial and highly specialized biological membranes, was further reported for various living cells from prokaryotes to yeast and mammalian cells [5,6,7]. The modulation of the fluidity of membrane domains is critical for cell function. For instance, the modulation of the fluidity of membrane domains can alter the distribution and function of membrane-bound receptors, enzymes, and other proteins diffusing laterally along the cell’s surface and intracellular organelle membranes [5]. The term membrane fluidity can thus be referred to as the degree of stiffness or rigidity of the cellular bilayers. Saturated fatty acid moieties in phospholipids are linear-chained and pack easily together or with the neighbor-cholesterol in the bilayer membrane. Unsaturated fatty acyl chains, on the other hand, retain bents along their long axis at the position of double bonds and therefore poorly aligned. Consequently, the high abundance of unsaturated fatty acids in membrane phospholipids increases the degree of membrane fluidity. As a whole, the physicochemical properties of the fatty acid moieties in membranes determine their function, and when altered it may impact the cells and organisms [5].

Fluorescent fatty acids and phospholipid derivatives provide the opportunity to follow the incorporation of free fatty acids both biochemically and morphologically during cellular development. Among the fluorescent fatty acids derivatives, one of the most important compounds for imaging is laurdan, composed of an acyl chain of lauric fatty acid (hydrophobic) with a 6-dimethylaminonaphthalene fragment (hydrophilic) linked at position 2 of the naphthalene ring [8]. Due to partial charge separation between the 6-dimethylamino and the 2-carbonyl residues, the naphthalene moiety has a dipole moment, which increases upon excitation and causes the reorientation of the surrounding solvent dipoles. One of its most important characteristics is its sensitivity to membrane phase transitions and other alterations to membrane fluidity such as the penetration of water [9,10,11,12,13,14]. The fluorescence spectrum of the probe laurdan, which incorporates into the lipid phase in the membrane, is correlated to its physical state. Two-photon infrared excitation techniques have been successfully applied to detect laurdan emission. Laurdan’s excited-state relaxation, independent of the head-group type in phospholipids, is highly sensitive to the presence and mobility of water molecules within the membrane bilayer, yielding information on membrane fluidity by a shift in its emission spectrum depending on the surrounding lipid phase state (i.e., bluish in ordered, gel phases and greenish in disordered, liquid-crystalline phases) [15]. In membranes, the excited-state relaxation depends on the number of the surrounding water molecules, and thus increases with membrane hydration levels [12,16,17,18,19]. This effect can also be revealed by a decrease of the fluorescence lifetime due to the enhanced emission from the relaxed state, which is characterized by a lower lifetime. By using this probe, coexisting lipid domains are distinguished based on their distinctive fluorescence lifetime (high membrane fluidity, low lifetime; low membrane fluidity, high lifetime). Fluorescence lifetime is independent of excitation intensities, probe concentrations, and other artifacts, relying on the ratiometric properties of the probe [6,20].

Here, we engineered novel laurdan-type fatty acid derivatives with increased acyl carbon chain length (SG12:0 (laurdan), SG14:0, SG16:0, and SG18:0), linked to the environmentally sensitive dimethylaminonaphthalene moiety. Fluorescence lifetime imaging analysis allowed us not only to monitor the intracellular distribution of fatty acids within the cell where the lipids accumulate, but also to examine how the fluidity and the microviscosity of the membrane environment influences their localization. Moreover, the extension of the environmental spectrum of phases, which are accessible to fatty acid derivatives with different chain length, allowed one to examine the effects of intracellular non-polar regions and acidic regions on the photophysics of the naphthalene moiety. The insights we obtained will pave the way to establish a method to monitor selectively lipophagy and related events.

## 2. Results

### 2.1. Synthesis of the Laurdan Analogs with Elongated Fatty Acid Chain

The synthetic route for the preparation of the laurdan counterparts SG14:0, SG16:0, and SG18:0 with different alkyl chain length is described in Figure 1. The overall route includes three synthetic steps. First, starting from commercially available fatty acids (myristic, palmitic, and stearic acids) and *N*,*O*-dimethylhydroxylamine hydrochloride the corresponding Weinreb amides 1–3 were prepared using EDC and DMAP as coupling reagents. In the second step, 6-bromo-*N*,*N*-dimethylnaphthalen-2-amine (4) was synthesized by reductive methylation of the commercially available 2-amino-6-bromonaphthalene with aqueous formaldehyde, sodium borohydride, and sulfuric acid. Finally, bromonaphthalene 4 was subjected to low-temperature bromine-to-lithium exchange with n-BuLi followed by the coupling with the Weinreb amides 1, 2 to give the laurdan derivatives; SG14:0 and SG16:0 respectively in 60–65% yield. Even higher yield of the analog SG18:0 (88%) was achieved upon an appropriate modification of the last step, namely using bromine-to-lithium in 4 followed by lithium-to-magnesium bromide metathesis with magnesium bromide etherate prior the coupling with the Weinreb amide 3. Such a metathesis with the formation of moderately basic naphthylmagnesium intermediate considerably decreased undesirable deprotonation of the amide 3, thus favoring its more efficient coupling with the ketone formation SG18:0.

### 2.2. FLIM Analysis of Fluorescence Analogs: Phase Behavior

In Figure 2 fluorescence imaging microscopy (FLIM) images of PC12 cells treated with laurdan (SG12:0) and the laurdan counterparts (SG14:0; SG16:0; SG18:0), integrated in the spectral range 400–600 nm, are reported. The first image (SG12:0) was pseudo-colored in a range going from red (low lifetime 2392 ps—fluid phase), indicated as $\tau^{fluid}$, to blue (high lifetime 5070 ps—gel phase), i.e., $\tau^{gel}$.

In the images and in the relative phasor plots it is possible to visualize and quantify the relative contributions of gel and fluid phases: in the SG12:0 image, the gel phase (blue pixels) is mainly localized along the cell borders (plasma membrane). From the phasor plot we argued that these pixels are characterized by monoexponential decay (the points are aligned on the universal circle—see the Materials and Methods) with a high lifetime. This lifetime, characterized by a monoexponential decay and associated with the gel-like phase, is indicated as $\tau_{mono}^{gel}$ with a value 5000 ps. On the contrary, cell interior is characterized by fluid pixels (green-colored pixels characterized by a single exponential decay) whose associated lifetime of 4000 ps is indicated as $\tau_{mono}^{fluid}$. Inside the cell, it is possible to distinguish another group of pixels, colored in yellow, which constitutes a family of spherical organelles. A single exponential decay instead characterizes these with a lower lifetime (3000 ps), which is indicated as $\tau_{mono}^{low}$.

By increasing the length of the carbon chain (from SG12:0 to SG18:0) it is possible to observe a general decrease of the extent of compartmentalization (due to the increased hydrophobicity) and preferential localization of the probe on peculiar organelles and organelle regions, which are barely visible with laurdan (SG12:0). In particular, the number of gel-phase pixels ($\tau_{mono}^{gel}$) progressively decreased, indicating, as a general trend, the preferential compartmentalization of the probe in fluid regions. Indeed, in SG16:0 and in SG18:0 the probes fail to compartmentalize on plasma membranes, which are environments characterized by low fluidity [9,21]. This process is also visible on the phasor plot, in which a transition of the pixels from a high lifetime region ($\tau_{mono}^{gel}$) to lower lifetimes ($\tau_{mono}^{fluid}$ and $\tau_{mono}^{low}$) was clearly detectable. Moreover, the long-chain probes appeared to compartmentalize in organelles that are instead poorly labeled with SG12:0 and were characterized by an abnormally low value of lifetime instead labeled with SG12:0 (orange pixels). These pixels had another particular feature: in the phasor plot it is possible to observe a cloud broadening in SG16:0, more pronounced in the SG18:0, which is indicative of the emergence of a fast component of the lifetime of the naphthalene moiety who was barely, if not measured in the cell environment labeled with SG12:0 or SG14:0 (please refer to Section 4.4). This indicates that a transition from another electronic state in the decay of the naphthalene moiety, which is visible in these organelles. Using multiexponential decay fittings allows one to fit the decays with two exponential decays, (0.6 ns and 3.0 ns in the SG18:0 image) and confirm that the fluorescent emission of the fluorophores come from two different excited states. In summary, these orange pixels are characterized by a two-exponential decay with a low average lifetime of 1800 ps, which is defined as ${\bar{\tau }}_{two-exp}^{low}$.

### 2.3. Spatial Localization of Membrane Phases through Phasor-Driven Segmentation

To detect where these different phases are in cells, we remapped selected regions in the phasor plot to the original fluorescence image, thus providing segmentation based on pixels with similar spectral properties (Figure 3). We limited for clarity the comparison to the more extreme values of chain length (SG12:0 and SG18:0). We observe that in SG12:0 the gel phase ($\tau_{mono}^{gel}$) was mainly localized on the plasma membrane (PM), while in SG18:0 the probes failed to compartmentalize on PM, and the gel phase was almost not detectable. Environments characterized by the fluid ($\tau_{mono}^{fluid}$) and the low lifetime phase ($\tau_{mono}^{low}$) were instead visible with both probes. However, the spatial distribution of these phases in SG18:0 was quite different from the SG12:0, showing how the domains in which the two FA analogs were colocalized were distributed differently throughout the cell. Looking at the intermediate and the fluid phase, we could observe that SG18:0 was localized in less compact domains with respect to SG12:0. The most peculiar behavior of SG18:0 was that it compartmentalized more strongly in regions characterized by a slow decay component of the lifetime that we denoted in the figure as ${\bar{\tau }}_{two-exp}^{low\;\left(1\right)}$ and ${\bar{\tau }}_{two-exp}^{low\;\left(2\right)}$, depending on both the lifetime value and the relative weight of the low-lifetime component. In particular, ${\bar{\tau }}_{two-exp}^{low\;\left(1\right)}$ shows a higher average lifetime and a lower weight of the low-lifetime component, while ${\bar{\tau }}_{two-exp}^{low\;\left(2\right)}$ was characterized by a lower average lifetime and a higher relative weight of the high-lifetime component. Both these regions were characterized by an almost spherical shape.

### 2.4. FLIM Analysis of Fluorescence Derivatives: Solvent Relaxation and Microviscosity

Another quantity that can be measured analyzing the fluorescence decay is the rate of solvent relaxation, obtained from measuring the rate of the spectral shift [14,22]. The speed of solvent relaxation is related to the rotational mobility of the water molecules within the membrane and is referred to as membrane microviscosity [23]. When the rate of spectral relaxation is in the picosecond scale, the temporal resolution of the time-correlated single photon counting devices is unable to resolve any changes. However, in very viscous environments the time scale of relaxation can increase up to the nanosecond time scale. In this context, the change in membrane polarity due to water hydration from the effects due to viscosity can be uncovered by analyzing the decay time in the green channel (emission 540/50 nm) [14,22]. In Figure 4A the decay from a sample region of the cell is reported for the blue channel, the green channel, and the whole spectrum. While the decay of the blue channel appears as a single exponential in the log-lin scale of the graph, the excited-state decay in the green channel presents two apparent decays: one is due to the decay of the standard fluorescence emission from the relaxed state, and the other is an apparent decay time due to the populating process from the locally excited state to the relaxed state. The two processes give rise to a visible non-exponential decay in pixels showing a very high value of microviscosity. The decay of the whole region was mostly dominated by the decay of the blue channel, characterized by a higher intensity, though the non-exponential decay is in this case barely noticeable. The phasor distribution integrated for SG12:0 cells (*n* = 15) is shown for the blue channel and the green channel (Figure 4B,C). While the center of mass of the phasor lay along the universal circle for the blue channel indicating an exponential decay, the phasor of the green channel lies outside of the universal circle for the control cells, suggesting a non-exponential decay. Upon increasing the saturated chain length, the point rotates towards the universal circle. We can thus conclude that the environment in which long tail fluorescent analogs compartmentalize is less viscous. Additionally, in the green channel, it is possible to observe that the cloud broadening in SG16:0 is more pronounced in the SG18:0, indicating that the fast component of the lifetime is also present in this wavelength range.

### 2.5. Colocalization Experiments

We performed colocalization experiments, intending to identify which organelles are those characterized by $\tau_{mono}^{low}$ and the intracellular compartments characterized by an abrupt decrease of lifetime (${\bar{\tau }}_{two-exp}^{low\;\left(1\right)}$ and ${\bar{\tau }}_{two-exp}^{low\;\left(2\right)}$).

Previous studies [24] lead us to believe that $\tau_{mono}^{low}$ is characteristic of apolar regions of cells. To investigate this, we labeled cells with SG18:0 and Nile Red, a solvatochromic probe with high selectivity for lipid droplets, whose emission is maximum at 590 nm [24,25,26].

The panel in Figure 5 shows the integrated fluorescence emission intensity of SG18:0 in the wavelength range 400–475 nm (Figure 5A), along with the lipid droplets (LD) overlapped mask in red (Figure 5B). This mask was obtained by integrating the fluorescence signal in the range 575–600 nm. An intensity-based threshold (2 percentile of the brightest pixels) was fixed to remove the background fluorescent signal after noise measurement. The mask of LD obtained was then applied to the SG18:0 lifetime emission image, thus identifying a subset of points in the phasor plot corresponding to LD lifetime (Figure 5C). The lifetime distribution of LD, obtained by applying the LD mask on the lifetime image, was represented in the graph in Figure 5D, which shows the normalized frequency of pixels (*y*-axis) characterized by a certain lifetime (*x*-axis). The mean value of LD lifetime distribution was calculated according to the equation τ_mean_ = ∑_i_(τ_i_∙f_i__norm_), where f_inorm_ is the normalized frequency of pixels characterized by the i-th lifetime value (τ_i_). The mean value of LD lifetime, obtained by averaging the lifetime distributions from *n* = 25 cells, is τ_mean_ = (2973 ± 26) ps. We could thus conclude that the pixels in the cell interior, characterized by a single exponential decay with $\tau_{mono}^{low}$ approximately (~) 3000 ps in Figure 3, were pixels emitting from the non-polar regions of LD.

We then investigated if the peculiar two-exponential decay could be due to the acidity of intracellular compartments. Therefore, we co-labeled cells with SG18:0 and LysoTracker RED dye, a probe that accumulates in acidic organelles [27].

A mask for acidic organelles was obtained by integrating the LysoTracker dye fluorescence signal in the emission range 575–600 nm and applying an intensity-based threshold (50 percentile of brightest pixels) to remove the background and isolate the contribution of acidic organelles after noise measurement (Figure 6A). We then provided a phasor-driven segmentation, to identify three regions of interest, indicated in Figure 6B as ROI 1, ROI 2, and ROI 3, respectively. These three regions differ in the weight of the low lifetime component, which increases in going from ROI 1 to ROI 3. Phasor-driven segmentations are shown in cyan (ROI 1), green (ROI 2), and magenta (ROI 3), respectively (Figure 6C–E), and the overlap is reported in Figure 6F. In Figure 6G, the percentage of the pixels labeled with SG18:0 was reported. The fraction of pixels decreased while going from the high lifetime region ROI 1 (68.7% ± 10.5%) to ROI 2 (6.6% ± 1.0%), with the minimum value detected in ROI 3 (0.9% ± 0.3%). On the contrary, the normalized emission intensity of LysoTracker associated with each ROI, reported in the graph in Figure 6H, and normalized to the fraction of bright pixels in the image, increased from 0.22 ± 0.02 in ROI 1 to 0.26 ± 0.02 in ROI 2 and it reached a maximum in ROI 3. Since LysoTracker compartmentalization is known to increase with compartment acidity [27], this result indicates that the weight of the low lifetime component is correlated to a pH increase.

## 3. Discussion

In this study, we constructed novel fatty acid derivatives with elongated aliphatic chain length assembled with the environmentally sensitive dimethylaminonaphthalene fragment. The fluorescence lifetime imaging analysis allowed us not only to monitor the compartmentalization of the long aliphatic tail within the cell where the lipids accumulate, but also to examine how the fluidity and the microviscosity of the membrane environment influences their localization.

The first significant result made was how the intracellular distribution of fatty acid fluorescent derivatives depended on the acyl chain length. The longer the acyl chain, the more fluid is the environment in which the probe preferentially portions. As a limit case, SG18:0 does not compartmentalize in plasma membranes ($\tau_{mono}^{gel}$ pixels), which are, as known, the less fluid regions of the membranes. This result clearly show that long acyl chains prefer membrane regions that are more hydrated ($\tau_{mono}^{fluid}$), thus less packed. Looking at the intermediate and the fluid phase, we can observe that SG18:0 is localized in less compact domains compared to SG12:0. Therefore, the preferential localization depends not only on the fluidity state monitored by naphthalene moiety, which is the hydrophilic–hydrophobic membrane interface, but also from other variables as the apparent viscosity and hydrophobicity of the side chain region (center) of membranes [28]. We also monitored concomitantly the microviscosity of the membrane environment, which depends on the rotational diffusional processes, which is strongly linked to the water–membrane interaction extent. The decrease in microviscosity observed with increasing FA chain length can be due to preferential compartmentalization of small chain FA with cholesterol-rich domains. Indeed, the increase of microviscosity is ascribed to the formation of water bridges between cholesterol and the phospholipid tails [6]. The cholesterol hydroxyl group can act both as a hydrogen bond donor and acceptor and also participate in charge pairing. That gives cholesterol the versatility to form numerous different types of bonds in the interfacial region. However, the increase in FA length may alter the dynamics of the formation of these bonds since cholesterol interactions are favored with acyl chains rather than water, as shown in several molecular dynamics (MD) and NMR studies on model membranes of DPPC and DMPC [29,30,31]. The overall increase of the extent of hydrogen-bonded water molecules with acyl tail can therefore explain the observed decrease in microviscosity with increasing FA chain length.

Peculiar behavior of all the synthesized laurdan analogs is their ability to cluster in very nonpolar compartments such as lipid droplets, as shown in colocalization experiments with the LD selective probe Nile Red [25], their lifetime is the lowest ($\tau_{mono}^{low}$), instead of being the one characteristic of less hydrated and thus rigid regions. $\tau_{mono}^{fluid}$ instead present a broader and more spotted distribution. The reason of this behavior can be explained as following (Figure 6): in addition to specific solvent-fluorophore effect allowing one to estimate membrane fluidity, the aminonaphthalene moiety can form an internal charge transfer state (ICT) [16]. In the case of aminonaphthalene-functionalized fatty acids, the fluorophore contains an electron-donating (the amino group) and an electron-accepting group (the carbonyl group). Part of the large spectral shift is due to emission from the locally excited (LE) state (lifetime 4–5 ns), which occurs near 400 nm, and from an ICT state emitting at longer wavelengths (3.8 ns). In the LE state the excitation is localized on the naphthalene ring, and the amino and the carbonyl groups are not part of the delocalized electron system [16]. In a highly nonpolar environment, where solvent dipoles’ alignment is prevented, the ICT state becomes the lowest energy state instead, with complete charge transfer from the amino group to the carbonyl group. Hence, the role of solvent polarity is not only to lower the energy of the LE state, but also to govern, which state has lower energy (Figure 5). A very low lifetime (less than 1000 ps) was already detected in the experiments with laurdan and prodan in strongly nonpolar solvents [32]. We have moreover observed that the most peculiar characteristic of the photophysics of long-chain fatty acids derivatives is their compartmentalization in regions characterized by a slow, two-exponential decay component of the average lifetime ${\bar{\tau }}_{two-exp}^{low\;\left(1\right)}$, resulting from the combination of two lifetimes being 0.6 ns and 3.0 ns in SG18:0 image, but this effect is barely detectable in living cells with short chain FA derivatives. We observed in colocalization experiments with LysoTracker that regions presenting both decays colocalize with acidic organelles. The lifetime of the longest component coincides with the one of the highly nonpolar ICT state, indicating that in lysosomes the ICT state is emitting. We speculate that low lifetime photons come from an energetically favored state at low pH, called pHE (pH-activated emission state), which emits along with ICT in lysosomes. We moreover observed that the weight of the low lifetime component seems to be pH-dependent, since we observed that LysoTracker compartmentalization, which is known to increase with compartment acidity, is higher, increasing the low lifetime weight.

In summary (Figure 7), the naphthalene functionalized FA derivatives present a locally excited, solvatochromic state coming from the probe localized on plasma membranes and inner membranes, an internal charge transfer (ICT) state coming from the probe localized in lipid droplets, and a two-state emission from ICT and pHE coming from the probe localized in lysosomes. In the highly nonpolar environment of LD, where the alignment of solvent dipoles is prevented, the ICT state becomes the lowest energy state, with complete charge transfer from the amino-group to the carbonyl group. In low pH environments of lysosomes, low lifetime photons come from pHE, an energetically favored state at low pH, which emits along with ICT.

We believe that these novel fatty acid/laurdan analogs with increasing saturated chain length (from 12 to 18 carbons), connected to the environmentally sensitive aminonaphthalene molecule, in conjunction with FLIM analysis allow not only to monitor the compartmentalization of fatty acids within the cell, but also to examine how the fluidity and the microviscosity of the membrane environment modulate their localization. Moreover, fatty acid derivatives with the longest chain compartmentalize in lipid droplets and lysosomes with characteristic lifetimes, thus making this probe a promising tool for monitoring lipophagy and related events. However, further investigation is needed to investigate and make more quantitative this interesting aspect. The analysis is in principle feasible, but not simple because the whole visible spectrum is required for analyzing the outcome of these probes. Other limitations of this method are the requirement of a spectral detector to acquire the whole emission spectrum with pixel resolution, and a staining protocol to detect the spectral shift that occurs with varying the degree of polarity of the environment. The resolution limit of 200 nm can be overcome, in principle, by the use of polarity-sensitive probes with super-resolution techniques [33].

## 4. Materials and Methods

### 4.1. Materials for the Synthesis of Laurdan Derivatives

Commercially available reagents and solvents were purchased from Sigma-Aldrich—Merck (Rehovot, Israel), Holand-Moran (Yehud, Israel), Acros Organic (Fair Lawn, New Jersey, USA), Alfa Aesar (Ward Hill, Massachusetts, USA), and Bio-Lab Ltd. (Jerusalem, Israel), and were used without any additional purification. Dry THF was obtained by the distillation from a boiled blue benzophenone ketyl under nitrogen. Analytical thin-layer chromatography (TLC) was carried on aluminum sheets precoated silica gel 60 F_254_ from Merck using UV absorption and alkaline potassium permanganate (KMnO_4_) physical absorption for visualization. Flash chromatography (FC) column was performed on Geduran silica gel (SiO_2_) 60 (230−400 mesh) from Merck Millipore (Darmstadt, Germany) or on SiliaFlash (SiO_2_) P60 (230−400 mesh) from Silicycle (Quebec, Canada). The ^1^H, ^13^C NMR, and 2D spectra were recorded at room temperature on a Bruker Advance NMR spectrometers (Vernon Hills, IL) operating at 300, 400, and 600 MHz for ^1^H-channel and were in accordance with the assigned structures. To facilitate signal assignment, in addition to 1D spectra, five different types of 2D-NMR correlation experiments were carried out: COSY (through-bond ^1^H-^1^H correlation), HMQC (one-bond ^1^H-^13^C correlation), HMBC (long-range ^1^H-^13^C correlation), and HMBC-N (through-bond ^1^H-^15^N correlation). The spectra are reported in ppm units (δ) and coupling constants (J) in Hertz. The chemical shifts are referenced to TMS ((δ_H_ and δ_C_ = 0 ppm). The samples were prepared by dissolving the synthesized compounds in CDCl_3_ (δ_H_ = 7.26 ppm) or acetone-d_6_ (δ_H_ = 2.05 ppm). The splitting pattern abbreviations are as follows: s, singlet; d, doublet; t, triplet; q, quartet; quint, quintet; m, unresolved multiplet; dd, doublet of doublet; td, triplet of doublet; and br, broad. High-resolution mass spectra (HRMS) were obtained on a QTOF instrument (Agilent, Santa Clara, CA, USA), using electrospray ionization (ESI) or atmospheric pressure chemical ionization (APCI).

### 4.2. Synthesis of Laurdan Analogs

General procedure for the synthesis of the Weinreb amides: To a stirred mixture of *N*,*O*-dimethylhydroxylamine (975.4 mg, 2 eq), DMAP (4-dimethylaminopyridine) (1.20 g, 2 eq) in dry dichloromethane (50 mL) a solution of an appropriate acid (1 eq) and *N*-(3-dimethylaminopropyl)-*N*′-ethylcarbodiimide hydrochloride (EDC-HCl) (4.8 g, 5 eq) were added at once as a solids. The reaction mixture was stirred at r.t. overnight. The reaction mixture was washed with 10% aq citric acid (10 mL) and extracted with CH_2_Cl_2_ (2 × 100 mL). The combined organic extract was washed with water (2 × 30 mL), saturated aq Na_2_CO_3_, brine (30 mL), dried over anhydrous Na_2_SO_4_, filtered, and evaporated under reduced pressure to afford a crude product. Subsequent flash chromatography (FC) on silica gel yielded the pure Weinreb amides **1**–**3**.

N-methoxy-N-methyltetradecanamide (1): The title product was obtained using myristic acid (1.10 g). The crude product was purified by FC (elution with 10% EtOAc in hexane) to obtain compound 1 as a colorless solid (1.10 g, 75%). R_f_ = 0.5 (hexane/EtOAc, 9:1); ^1^H NMR (300 MHz, CDCl_3_): δ 3.68 (s, 3H), 3.18 (s, 3H), 2.41 (t, J = 7.5 Hz, 2H), 1.63 (quint, J = 7.2 Hz, 2H), 1.30–1.26 (m, 20 H), 0.88 (t, J = 6.8 Hz, 3H) ppm; ^13^C-NMR (75 MHz, CDCl_3_): δ 174.84, 61.20, 32.21 (br), 31.94, 29.67 (5C), 29.54, 29.49, 29.46, 29.38, 24.69, 22.70, 14.13 ppm; HRMS (ESI): m/z calcd for C_16_H_34_NO_2_^+^: 272.2584 [M+H]^+^; found: 272.2594. The NMR and MS corresponded to the previously reported data [34].

*N*-methoxy-*N*-methylpalmitamide (2): The title product was prepared from palmitic acid (1.30 g). The crude product was purified by FC (elution with 20% EtOAc in hexane) to give 2 as a colorless solid (1.20 g, 80%). Rf = 0.5 (hexane/EtOAc, 8:2); the NMR and MS results correlates with previous results reported [35]. ^1^H NMR (400 MHz, CDCl_3_): δ 3.67 (s, 3H), 3.16 (s, 3H), 2.41 (t, J = 7.4 Hz, 2H), 1.66–1.59 (m, 2H), 1.31–1.26 (m, 24 H), 0.88 (t, J = 6.8 Hz, 3H) ppm; ^13^C-NMR (100 MHz, CDCl_3_): δ 174.57, 60.93, 31.97 (br), 31.80, 31.74, 29.55 (7C), 29.39, 29.32, 29.24, 24.29, 22.54, 13.92 ppm; HRMS (ESI): m/z calcd for C_18_H_38_NO_2_^+^: 300.2897 [M+H]^+^; found: 300.2905.

N-methoxy-N-methylstearamide (3): The title product 3 (0.86 g, 52%) was prepared using stearic acid (1.30 g) and purified by FC (elution with 50% EtOAc in hexane). ^1^H NMR (300 MHz, acetone-d_6_): δ 3.70 (s, 3H), 3.10 (s, 3H), 2.39 (t, J = 7.4 Hz, 2H), 1.57 (quint, J = 7.4 Hz, 2H), 1.29 (m, 28 H), 0.88 (t, J = 7.4 Hz, 3H) ppm. ^1^H NMR spectrum of 3 matched the previously reported data [36].

Synthesis of 6-bromo-*N*,*N*-dimethylnaphthalen-2-amine (4): The compound was synthesized following the described protocol [37]. Briefly, to a stirred mixture of 3M sulfuric acid (3.99 mL, 2.5 eq) and 40% aqueous formaldehyde (2.3 mL, 6 eq) at 0 °C a solution of 2-amino-6-bromonaphthalene (1.11 g, 1 eq) and sodium borohydride (1.32 g, 7 eq) in THF (35 mL) was added dropwise. After the addition, the pH was tested and kept around 6.0. The reaction mixture was stirred for 2 h and quenched with NaOH pellets until the solution became basic (pH = 10). The supernatant solution was decanted, the solid residue was dissolved in water (100 mL) and extracted with dichloromethane (2 × 100 mL). The combined organic extract was washed with brine (50 mL), dried over anhydrous Na_2_SO_4_ and filtered. The solvent was evaporated under reduced pressure to afford a crude product 4 as an orange solid (1.17 g, 94%), m.p = 120 °C; R_f_ = 0.8 (hexane/EtOAc 8:2). ^1^H NMR (300 MHz, CDCl_3_): δ 7.81 (d, J = 1.8 Hz, 1H), 7.58 (d, J = 9.1 Hz, 1H), 7.50 (d, J = 8.8 Hz, 1H), 7.39 (dd, J = 8.8, 2.0 Hz, 1H), 7.14 (dd, J = 9.1, 2.6 Hz, 1H), 6.84 (d, J = 2.5 Hz, 1H), 3.02 (s, 6H) ppm; ^13^C NMR (75 MHz, CDCl_3_): δ 149.05, 133.76, 129.66, 128.14, 128.11, 128.03, 117.36, 115.31, 106.34, 41.00 ppm; HRMS (ESI): m/z calcd for C_12_H_13_BrN^+^: 250.0226 [M+H]^+^; found: 250.0231. The product was used in the next steps without any purification.

General procedure for synthesis of the laurdan analogs SG14:0, SG16:0, and SG18:0. To a cold (−78 °C) solution of bromonaphthalene 4 (500 mg, 1.2 eq) and dry THF (10 mL) under nitrogen a solution n-butyllithium in hexanes (2.5 M, 1.0 mL, 1.4 eq) was added slowly. The reaction mixture was stirred at −78 °C for 1 h to give a yellow suspension, which was allowed to warm spontaneously to 0 °C over 2 h. At that time, a solution of the Weinreb amide 1, 2 (1 eq) in THF (3 mL) was added. The reaction mixture was stirred at room temperature overnight and quenched by addition of H_2_O (10 mL). The mixture was diluted with water (100 mL) and extracted with dichloromethane (3 × 100 mL). The combined organic extract was washed with brine (50 mL), dried over anhydrous Na_2_SO_4_, filtered, and concentrated under reduced pressure. The residue was purified by FC to give pure laurdan analogs SG14:0 and SG16:0 respectively.

1. -(6-(Dimethylamino)naphthalen-2-yl)tetradecan-1-one (SG14:0): The title product was synthesized using the Weinreb amide 1 (0.45 g). The crude product was purified by FC (gradient elution: from 2% to 10% EtOAc in hexane) give SG14:0 as a yellow solid (415 mg, 65%). m.p. 74 °C, R_f_ = 0.6 (hexane/EtOAc 9:1). ^1^H NMR (400 MHz, acetone-d_6_): δ 8.44 (br s, 1H), 7.91–7.88 (m, 2H), 7.68 (d, J = 8.7 Hz, 1H), 7.28 (dd, J = 9.1, 2.6 Hz, 1H), 6.99 (d, J = 2.6 Hz, 1H), 3.11 (s, 6H), 3.06 (quint, J = 7.2 Hz, 2H), 1.74 (quint, J = 7.3 Hz, 2H), 1.42–1.28 (m, 20H), 0.88 (t, J = 7.3 Hz, 3H) ppm; ^13^C-NMR (100 MHz, acetone-d6): δ 199.54, 151.37, 138.59, 131.63, 131.47, 130.58, 126.93, 126.14, 125.15, 117.28, 106.06, 40.47, 38.60, 32.65, 30.41, 30.31, 30.29, 30.15, 30.07, 25.45, 23.33, 14.35 ppm; HRMS (ESI): m/z calcd for C_26_H_40_NO^+^: 382.3104 [M+H]^+^; found: 382.3105.

1. -(6-(Dimethylamino)naphthalen-2-yl)hexadecan-1-one (SG16:0): The title laurdan analog SG16:0 (400 mg, 60%) was synthesized according to the General Procedure from the Weinreb amide 2 (0.50 g) and purified by FC (elution with 5% EtOAc in hexane) followed by recrystallization from hexane. A yellow solid, m.p. 83 °C, R_f_ = 0.4 (hexane/EtOAc 9:1). ^1^H NMR (600 MHz, acetone-d_6_): δ 8.43 (d, J = 0.9 Hz, 1H), 7.90 (dd, J = 8.6, 1.7 Hz, 1H), 7.89 (d, J = 9.0 Hz, 1H), 7.68 (d, J = 8.6 Hz, 1H), 7.28 (dd, J = 9, 2.6 Hz, 1H), 6.99 (d, J = 2.6 Hz, 1H), 3.11 (s, 6H), 3.07 (t, J = 7.4 Hz, 2H), 1.74 (quint, J = 7.4 Hz, 2H), 1.44–1.35 (m, 4H), 1.30–1.28 (m, 22H), 0.88 (t, J = 7.4 Hz, 3H) ppm; ^13^C-NMR (150 MHz, acetone-d_6_): δ 199.52, 151.35, 138.58, 131.62, 131.41, 130.53, 126.93, 126.13, 125.09, 117.27, 106.06, 40.47, 38.60, 32.65, 30.41, 30.37, 30.31, 30.29, 30.14, 29.97, 29.84, 29.71, 29.58, 25.44, 23.33, 14.35 ppm; HRMS (APCI): m/z calcd for C_28_H_44_NO^+^: 410.3417 [M+H]^+^; found: 410.3418.

1. -(6-(Dimethylamino)naphthalen-2-yl)octadecan-1-one (SG18:0): To a cold (−78 °C) solution of bromonaphthalene 4 (300 mg, 1.2 eq) and dry THF (10 mL) under nitrogen, n-butyllithium in hexanes (2.5 M, 1.0 mL, 1.4 eq) and a solution of magnesium bromide ethyl etherate (0.37 gr, 1.4 eq) in dry THF (2 mL) were carefully added. The reaction mixture was stirred at −78 °C for 1 h, which was allowed to warm spontaneously to 0 °C over 2 h. At that time, a solution of the Weinreb amide 3 (0.36 gr, 1 eq) in THF (3 mL) was added. The reaction mixture was stirred at room temperature overnight and quenched by addition of 2 M aqueous NH_4_Cl (5.0 mL). The mixture was diluted with water (100 mL) and extracted with a mixture of EtOAc:hexane (1:1, 2 × 150 mL). The combined organic extract was washed with brine (50 mL), dried over anhydrous Na_2_SO_4_, filtered, and concentrated under reduced pressure. The residue was purified by FC on silica gel (elution from 20 to 50% toluene in hexane) to give pure laurdan analog SG18:0 as a yellow solid. (463.7 mg, 88%). m.p. 82 °C, R_f_ = 0.5 (hexane/EtOAc 8:2); ^1^H NMR (400 MHz, CDCl_3_): δ 8.24 (s, 1H), 7.85 (dd, J = 8.8, 1.4 Hz, 1H), 7.72 (d, J = 8.9 Hz, 1H), 7.56 (d, J = 8.8 Hz, 1H), 7.09 (dd, J = 8.9, 2.4 Hz, 1H), 6.81 (d, J = 1.5 Hz, 1H), 3.03 (s, 6H), 2.95 (t, J = 7.4 Hz, 2H), 1.70 (quint, J = 7.4 Hz, 2H), 1.34–1.18 (m, 28H), 0.81 (t, J = 7.6 Hz, 3H) ppm; ^13^C NMR (100 MHz, CDCl_3_): δ 199.2, 149.1, 136.5, 129.8, 129.7, 128.7, 125.2, 124.2, 123.7, 115.3, 104.5, 39.5, 37.4, 30.9, 28.7–28.4 (13 CH2), 21.7, 13.1 ppm; HRMS (ESI): m/z calcd for C_30_H_48_NO^+^: 438.3730 [M+H]^+^; found: 438.3743.

A complete characterization of the synthesized compounds is provided in Appendix A.

### 4.3. Cell Labeling and Fluorescence Lifetime Imaging Microscopy Acquisitions

PC12 rat pheochromocytoma cells (ATCC, Manassas, VA, USA) were maintained in standard RPMI 1640 medium (Thermo Fisher Scientific, Waltham, MA, USA) supplemented with 10% fetal calf serum (FCS), 1 mM sodium pyruvate, and 10 mM HEPES (Sigma-Aldrich, St. Louis, MO, USA, dilution 1:100), Pen/Strep (10,000 U/mL penicillin and 10,000 μg/mL streptomycin, Gibco, USA, dilution 1:100), and 50 μM β-Mercaptoethanol (Gibco, Gaithersburg, MD, USA, dilution 1:1000). For FLIM analysis, cells were seeded in 35-mm Petri dishes (Greiner Bio-One International GmbH, Kremsmünster, Austria) at a density of 3 × 105 cells/cm^2^ and incubated overnight to facilitate cell adhesion.

The stock solutions of 1 mM Nile Red and Laurdan (Thermo Fisher Scientific, Waltham, MA, USA) and other synthesized fatty acids analogs SG14:0, SG16:0, and SG18:0, respectively, were prepared in DMSO (Sigma-Aldrich, St. Louis, MO, USA). Cells, stained with a 1 µM solution of different probes, were incubated for 24 h in the dark, to ensure compartmentalization. For colocalization experiments, the 1 mM LysoTracker RED stock solution (Thermo Fisher Scientific, Waltham, MA, USA) was diluted to 50 nM final working concentration in the growth medium. Cells were incubated with the prewarmed (37 °C) probe-containing medium for 1 h and then the loading solution was replaced with fresh medium.

FLIM (fluorescence imaging microscopy) data were acquired with a confocal microscope Nikon A1-MP (Nikon Corporation, Minato, Tokyo, Japan) equipped with an on-stage incubator (37 °C, 5% CO_2_, Okolab, Pozzuoli, NA, Italy) and a 2-photon Ti:Sapphire laser (Mai Tai, Spectra Physics, Newport Beach, CA) producing 80-fs pulses at a repetition rate of 80 MHz. A PML-SPEC 16 GaAsP (bandwidth 12.5nm, wavelength range 400–600 nm, Becker & Hickl GmbH, Berlin, Germany) multiwavelength detector coupled to a SPC-830 TCSPC/FLIM device (Becker & Hickl GmbH, Berlin, Germany) was used to collect the decay data. Laurdan excitation, set at 780 nm, induced negligible autofluorescence, since two-photon excitation of intracellular metabolites was centered at 740 nm. A 60× oil-immersion objective, 1.2 NA, was used for all experiments. For image acquisition, the pixel frame size was set to 512 × 512 and the pixel dwell time was 60 μs. The average laser power at the sample was maintained at the mW level.

### 4.4. Phasor Analysis

Phasor analysis is a widely used method based on the Fourier transform representation of a given curve [38]. FLIM acquisitions provide for the decay curves, *I(t)*, for each pixel. Two coordinates, g corresponding to the real part and s corresponding to the imaginary part, respectively, can be calculated according to the following equations:(1)$$g\left(\omega \right)=\frac{\int_{0}^{\infty }I\left(t\right)cos\left(\omega t\right)}{\int_{0}^{\infty }I\left(t\right)}$$
(2)$$s\left(\omega \right)=\frac{\int_{0}^{\infty }I\left(t\right)sin\left(\omega t\right)}{\int_{0}^{\infty }I\left(t\right)}$$
where ω is the laser repetition angular frequency, obtained by multiplying the laser repetition rate by 2π. These two coordinates individuate a single point, called phasor, in a scatter plot, which is known as the phasor plot.

Coincident decays are projected on the same point, while different decays are projected on different points. This technique allows detecting and quantifying changes in decay times without the requirement of an explicit mathematical model, thus constituting an ideal model-free approach. According to the phasor rules, a phasor point or population lying or centering on the universal semicircle indicates a single-lifetime species. On the contrary, the phasor of a complex species is a linear combination of the individual phasors of single-lifetime species. Connecting these individual phasors on the semicircle yields a convex set inside the semicircle confines species of multiple lifetime components inside the semicircle. However, when a fluorescent component is an excited-state product, its phasor would traverse outside the semicircle [6,39].

Depending on the position on the phasor plot, the phase state of the environment and the type of decay associated with each lifetime were indicated. The phase state, identified by the apex, can be gel-like or liquid-like (gel and liquid, respectively), while the pedex refers to exponential decay (mono for the single-lifetime species and two-exp, for the complex species, respectively).

An image containing several different decay traces will result in a cloud of point scattered throughout the plane. Selected regions of interest (ROI) in the phasor plot can be remapped to the original fluorescence image, thus providing segmentation based on pixels with similar lifetime decays. Lifetime phasors analysis was performed through the PhasorM software [24,25,26].

### 4.5. Statistics

Statistical analysis for sets of biological/biophysical data was performed by R Studio [40]. Mean values ± standard error were reported. Baseline characteristics were compared with the *t*-test for parametric variables and with ANOVA and a Tukey test when more than two groups were involved.

## Figures and Tables

**Figure 1 ijms-22-03106-f001:**
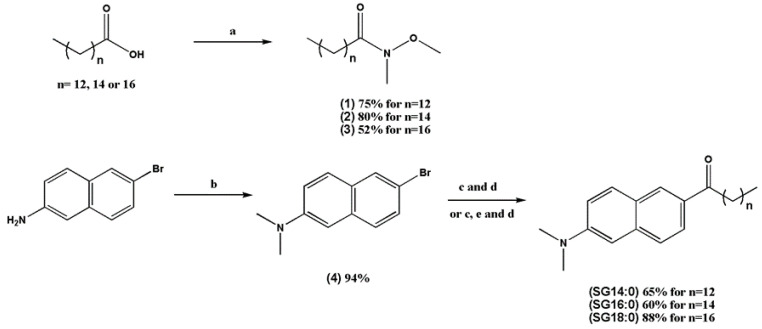
Synthesis of SG14:0, SG16:0, and SG18:0. Reagents and conditions: (**a**) N, O-dimethylhydroxylamine hydrochloride, EDC-HCl, DMAP in CH_2_Cl_2_, rt, 12 h. (**b**) 3 M H_2_SO_4_, 40% aqueous formaldehyde, NaBH_4_, THF, pH = 6, from 0 °C to rt, 2 h. (**c**) *n*-Buli, THF, −78 °C, 1 h, from −78 to 0 °C, 2 h. (**d**) Weinreb amide **1**–**3**, THF, 0 °C 12 h. (**e**) MgBr_2_ etherate, THF, −78–0 °C, 2 h.

**Figure 2 ijms-22-03106-f002:**
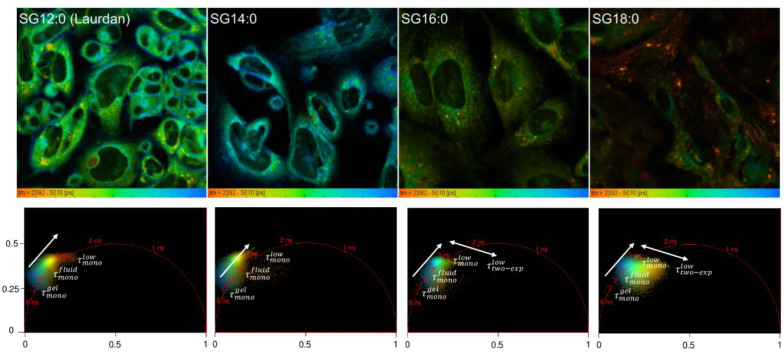
Whole Spectrum fluorescence imaging microscopy (FLIM) analysis of PC-12 labeled with fatty acids fluorescent derivatives. FLIM images of PC12 cells treated with laurdan (SG12:0) and the laurdan analogs (SG14:0; SG16:0; and SG18:0), integrated in the spectral range 400–600 nm, are reported. Images (first row) and phasor plot pixels (second row) are pseudo-colored in a range going from red (low lifetime 2500 ps—fluid phase) to blue (high lifetime 5000 ps—gel phase).

**Figure 3 ijms-22-03106-f003:**
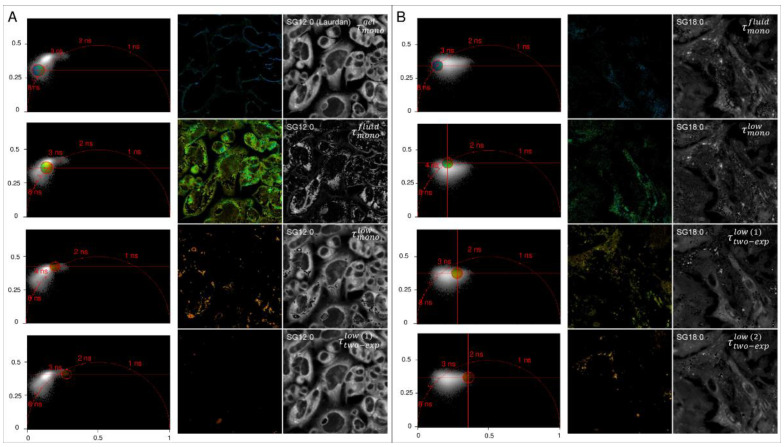
Phasor-driven segmentation of fatty-acid derivatives for SG12:0 (**A**) and SG16:0 (**B**). In the first column the phasor plot of a representative SG12:0 image is reported. On the phasor plot, selected pixels in a circular Region-of-Interest (ROI) are remapped to the original fluorescence image. In the second column the original image without the segmented phasor pixel is reported, and in the third column the phasor-driven segmentation is reported. In SG12:0 the gel phase ($\tau_{mono}^{gel}$) is mainly localized on the PM, while in the SG18:0 the probe failed to compartmentalize on PM, and the gel phase is almost no detectable. Environments characterized by intermediate ($\tau_{mono}^{fluid}$) and fluid phase ($\tau_{mono}^{low}$) are instead visible with both probes.

**Figure 4 ijms-22-03106-f004:**
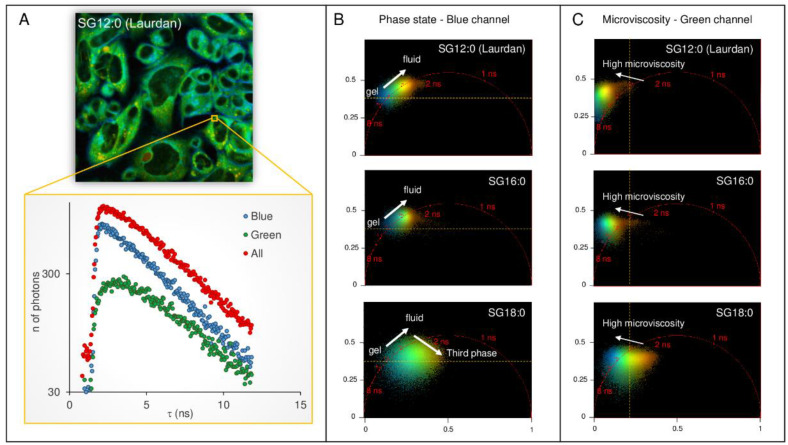
Spectrally resolved phasor analysis of fatty acid derivatives fluorescence. (**A**) The decay from a sample region of the cell is reported for the blue channel, the green channel, and whole spectrum. While the decay of the blue channel appears as a single exponential in the log-lin scale of the graph, the excited-state decay in the green channel presents two apparent decays: one is due to the decay of the standard fluorescence emission from the relaxed state, and the other is an apparent decay time due to the populating process from the locally excited state to the relaxed state. The two processes give rise to a visible non-exponential decay in pixels showing a very high value of microviscosity. The phasor distribution integrated for SG12:0 cells (*n* = 15) is shown for the (**B**) blue channel and the (**C**) green channel. While the center of mass of the phasor lies along the universal circle for the blue channel indicating exponential decay, the phasor of the green channel lies outside of the universal circle for the control cells, indicating a non-exponential decay. Upon increasing of the aliphatic chain length, the point rotates towards the universal circle.

**Figure 5 ijms-22-03106-f005:**
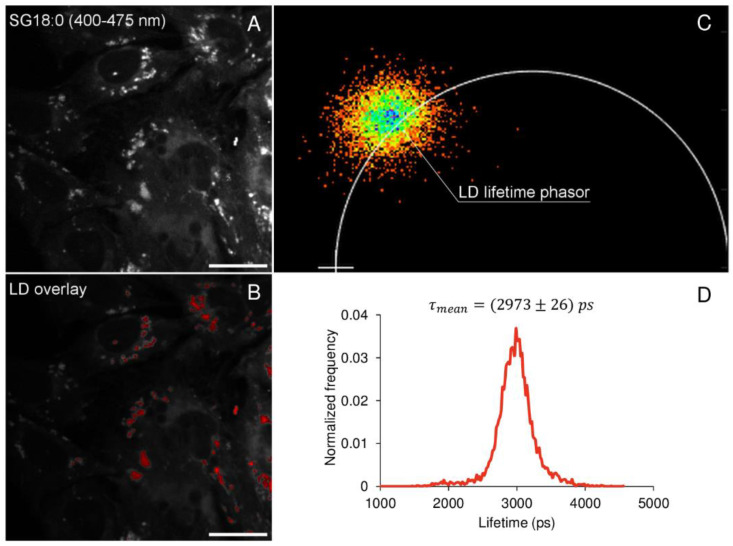
Nile Red colocalization experiment: lifetime distribution of lipid droplets. (**A**) The integrated fluorescence emission intensity of SG18:0 in the wavelength range 400–475 nm. In (**B**) the mask of lipid droplets, obtained by integrating the Nile Red fluorescence emission between 575 and 600 nm and taking into account the 2 percentile of the brightest pixels to remove background overlapped to the image (red). The application of this mask to the SG18:0 lifetime emission image identifies the subset of points in the phasor plot that corresponds to the LD lifetime (**C**). The graph in (**D**) shows the lifetime distribution of LD, reported as the normalized frequency of pixels (*y*-axis) characterized by a certain lifetime (*x*-axis). The mean value ± standard deviation of LD lifetime was obtained by averaging the lifetime distribution from *n* = 25 cells.

**Figure 6 ijms-22-03106-f006:**
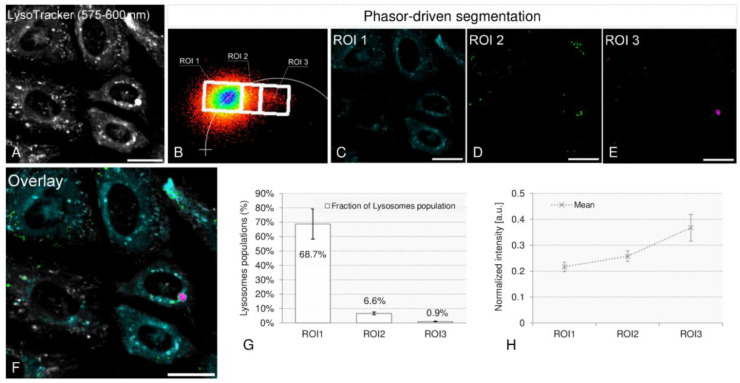
LysoTracker colocalization experiment: lifetime-based segmentation of lysosomes. (**A**) Mask for acid organelles obtained by integrating the LysoTracker fluorescence signal in the emission range 575–600 nm and by applying an intensity-based threshold (50 percentile of brightest pixels) to remove the background. A phasor-driven segmentation allows identifying three regions of interest, with decreasing lifetime, indicated in (**B**) as ROI 1, ROI2, and ROI 3, respectively. Phasor-driven segmentations are shown in cyan (ROI 1), green (ROI 2), and magenta (ROI 3), in (**C**–**E**), respectively, while the overlap is represented in (**F**). The graph in (**G**) shows the fractions, expressed as percentage, of each ROI’s pixels labeled with SG18:0. In (**H**) the normalized emission intensity of LysoTracker associated with each ROI and normalized to the fraction of bright pixels in the image is reported. Values, indicated as mean ± standard error, were evaluated on *n* = 25 cells.

**Figure 7 ijms-22-03106-f007:**
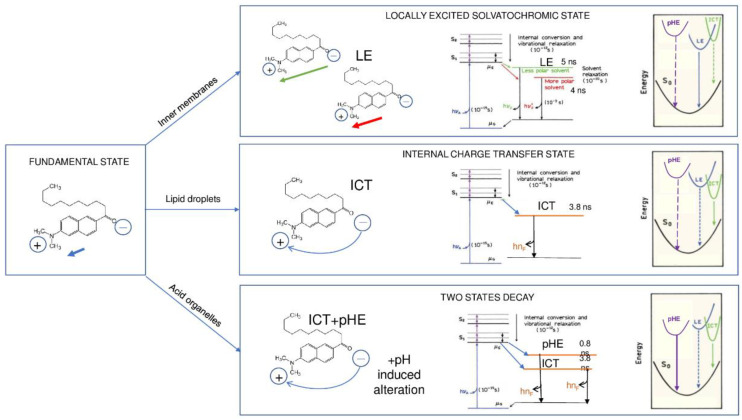
Photochemistry of dimethylaminonaphtalene-functionalized derivatives of fatty acids. The naphthalene functionalized FA derivatives present a locally excited, solvatochromic state coming from the probe localized on plasma membranes and inner membranes, an internal charge transfer (ICT) state coming from the probe localized in lipid droplets, and a two-state emission from ICT pHE (pH-activated emission state) coming from the probe localized in lysosomes. In highly nonpolar environment of LD, where the alignment of solvent dipoles is prevented, the ICT state becomes the lowest energy state, with complete charge transfer from the amino-group to the carbonyl group. In low pH environments of lysosomes, low lifetime photons come from an energetically favored state at low pH, called pHE, which emits along with ICT.

## Data Availability

Data are contained within the article. Additional data are available on request from the corresponding author.

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
