# Peer review of "Investigation of the Membrane Fluidity Regulation of Fatty Acid Intracellular Distribution by Fluorescence Lifetime Imaging of Novel Polarity Sensitive Fluorescent Derivatives"

_ijms, 2021, doi:10.3390/ijms22063106_

Round 1

Reviewer 1 Report

The article by Bianchetti et al. studies the membrane fluidity regulation of fatty acid distribution by FLIM of novel polarity sensitive fluorescent derivatives. The authors synthesized a series of fatty acid derivatives (12 to 18 carbons) conjugated to dimethylaminonaphtalene moiety (as in laurdan probe). FLIM analysis was performed in PC-12 cells and fatty acids derivatives with increased chain length were observed to compartmentalize in the ‘fluid regions’. Lipid droplets and lysosomes were shown to accumulate the derivatives with the longest chains. In general, the topic is interesting and the authors potentially provide new fluorescent tools, however, there are several drawbacks of the article, as following:

  1. In the introduction, the authors provide quite general information on fatty acids and lipids and cell membranes are described only from the perspective of ‘fluidity’. The concepts of membrane lateral heterogeneity, domain existence (such as the raft hypothesis) is not even mentioned. All the spectroscopic results are discussed only in the light of fluidity and microviscosity.
  2. It would be of great importance to characterize those derivatives in liposomes of a known and controlled composition before cellular measurements. The spectroscopic parameters could be discussed and understood in much more details.
  3. The fitting routine applied in FLIM is not properly described. All the parameters such as tau^{high}, tau^{low}, tau^{fluid} should be described in details somewhere (even in the methods section the description is not complete enough).
  4. Figs. 3 & 4 The numbers are not readable.

Author Response

Reviewer 1

The article by Bianchetti et al. studies the membrane fluidity regulation of fatty acid distribution by FLIM of novel polarity sensitive fluorescent derivatives. The authors synthesized a series of fatty acid derivatives (12 to 18 carbons) conjugated to dimethylaminonaphtalene moiety (as in laurdan probe). FLIM analysis was performed in PC-12 cells and fatty acids derivatives with increased chain length were observed to compartmentalize in the ‘fluid regions’. Lipid droplets and lysosomes were shown to accumulate the derivatives with the longest chains. In general, the topic is interesting and the authors potentially provide new fluorescent tools, however, there are several drawbacks of the article, as following:

1. In the introduction, the authors provide quite general information on fatty acids and lipids and cell membranes are described only from the perspective of ‘fluidity’. The concepts of membrane lateral heterogeneity, domain existence (such as the raft hypothesis) is not even mentioned. All the spectroscopic results are discussed only in the light of fluidity and microviscosity.

We thank the reviewer for pointing out this issue. Membrane phase equilibrium and heterogeneity is strictly related to membrane fluidity. We added these missing parts to the introduction (p. 2, lines 48-62).

2. It would be of great importance to characterize those derivatives in liposomes of a known and controlled composition before cellular measurements. The spectroscopic parameters could be discussed and understood in much more details.

Model membrane systems have aided in understanding the lipid organization of cell membranes. Planar supported bilayers, giant unilamellar vesicles (GUVs) made from lipid mixtures as well as giant plasma membrane (PM) vesicles (GPMVs) and PM spheres have provided views to visualize membrane domains, as for example reviewed in [1]. We agree with the reviewer that the ability to control the membrane composition of these model systems can be advantageous because it enables understanding the functional roles of specific lipids in the formation of domains. These were already deeply explored with the C12 moiety (laurdan) in several publications [2–4]. We will plan for sure another research study for investigating these model systems with the developed derivatives. However, since, according to literature, GUVs are less useful to extrapolate observations to systems with higher lipid compositional complexity such as the PM (in which also proteins are embedded) [1], we expect that the generalization of the results of such investigation to those obtained in this manuscript will be of a marginal entity. Striking differences in lipid or protein partitioning can even be found between GUVs and GPMVs [5].  

3. The fitting routine applied in FLIM is not properly described. All the parameters such as tau^{high}, tau^{low}, tau^{fluid} should be described in details somewhere (even in the methods section the description is not complete enough).

We thank the reviewer for pointing out this issue. We added a description of the parameters in the manuscript (see sections 2.2 and 2.3 of the Results, p. 4-6, lines 146-209). Moreover, we provided a more detailed explanation of the phasor analysis applied in the Materials and Methods section (p. 15-16, lines 579-611).

4. Figs. 3 & 4 The numbers are not readable.

We thank the reviewer for this observation. We added readable numbers in Figs. 3 and 4.

In addition, we corrected several typos to improve the readability of the manuscript.

References

[1] M. Carquin, L. D’Auria, H. Pollet, E.R. Bongarzone, D. Tyteca, Recent progress on lipid lateral heterogeneity in plasma membranes: From rafts to submicrometric domains, Prog. Lipid Res. 62 (2016) 1–24. https://doi.org/10.1016/j.plipres.2015.12.004

[2] L.A. Bagatolli, E. Gratton, Two photon fluorescence microscopy of coexisting lipid domains in giant unilamellar vesicles of binary phospholipid mixtures, Biophys. J. 78 (2000) 290–305. https://doi.org/10.1016/S0006-3495(00)76592-1

[3] L.A. Bagatolli, T. Parasassi, G.D. Fidelio, E. Gratton, A model for the interaction of 6-lauroyl-2-(N,N-dimethylamino)naphthalene with lipid environments: implications for spectral properties., Photochem. Photobiol. 70 (1999) 557–64. https://doi.org/10.1111/j.1751-1097.1999.tb08251.x

[4] L. Bagatolli, E. Gratton, T.K. Khan, P.L.G. Chong, Two-photon fluorescence microscopy studies of bipolar tetraether giant liposomes from thermoacidophilic archaebacteria Sulfolobus acidocaldarius, Biophys. J. 79 (2000) 416–425. https://doi.org/10.1016/S0006-3495(00)76303-X

[5] P. Sengupta, A. Hammond, D. Holowka, B. Baird, Structural determinants for partitioning of lipids and proteins between coexisting fluid phases in giant plasma membrane vesicles, Biochim. Biophys. Acta - Biomembr. 1778 (2008) 20–32. https://doi.org/10.1016/j.bbamem.2007.08.028

Reviewer 2 Report

I read with much interest the article entitled: "Investigation of the membrane fluidity regulation of fatty acid intracellular distribution by fluorescence lifetime imaging of novel polarity sensitive fluorescent derivatives" by Giada Bianchetti  et al. The Authors proposed novel laurdan-type fatty acid derivatives with increased acyl carbon chain length (SG12:0 [laurdan], SG14:0, SG16:0, SG18:0), linked to an environmentally sensitive dimethylaminonaphthalene moiety. They used fluorescence lifetime imaging analysis not only to monitor the intracellular distribution of fatty acids within the cell where the lipids accumulate but also to examine how the fluidity and the microviscosity of the membrane environment influence their localization. In addition, the extension of the environmental spectrum of phases accessible to fatty acid derivatives with different chain length, allowed to examine effects of intracellular non-polar regions and acidic regions on the photophysics of the naphthalene moiety.

The manuscript is well structured and contained with up-to-date references. I am convinced that it will be of potential interest to the science community. According to my opinion, the manuscript can be considered for publication.

Author Response

We thank the reviewer for the positive comments.

We corrected several typos to improve the readability of the manuscript.

Round 2

Reviewer 1 Report

The authors have implemented most of the Reviewer's suggestions. However the tick labels in the Fig. 2 still require improvement.

Author Response

We thank the Reviewer for pointing out this issue. According to the Reviewer's suggestion, we improved tick labels in Figure 2.